# Prenatal and postnatal droughts interact in shaping cognitive development
Fabienne Pradella [1,2,3,6] ✉, Sabine Gabrysch [2,4,5] & Reyn van Ewijk [1]

## Abstract

**Background** Developmental plasticity refers to biological adaptations, most often prenatally, to environmental cues. These can help organisms adapt to similar postnatal environments, with health benefits if prenatal and postnatal conditions match. While associations between various prenatal exposures and adverse offspring health have been documented, the interaction between prenatal and postnatal conditions remains less understood. We address this gap by examining whether pre- and postnatal drought exposures interact in their impact on cognitive performance, as early-life nutrition is a critical factor for cognitive development.

**Methods** Standardized math and reading scores from 11–16 year-olds in rural India ($N = 2,032,917$) from the 2007–2018 Annual Status of Education Report (a cross-sectional cognitive assessment household survey) were combined with University of Delaware Terrestrial Precipitation data. Given the high reliance on rainfed agriculture in the setting, rainfall levels below the 20th percentile of the district-specific long-term mean served as a proxy for nutritional adversities in a quasi-experimental study setup.

**Results** We show that early-life droughts adversely impact cognitive function. We find positive interaction terms between prenatal and postnatal drought exposures, suggesting that children already exposed to droughts prenatally are better equipped for postnatal droughts.

**Conclusions** The findings of this study align with the predictions around phenotypic plasticity, i.e., that prenatal conditions prepare organisms for similar postnatal challenges. However, given the increasing unpredictability of the climate, such alignments cannot be planned or anticipated, implying frequent mismatches between prenatal and postnatal conditions.

## Plain language summary

Pregnancy and the first years of life are critical periods for brain development, and adequate nutrition is essential during these times. In many regions, agriculture relies on rainfall, and droughts can cause nutritional shortages. We analyzed math and reading test results of two million children aged 11–16 years in rural India to understand how nutrition before and after birth affects later-life learning. We found that the combination of these two periods matters: droughts in early life generally lead to worse test results, but a drought during pregnancy cushions the effect of a drought a few years later. This is in line with the concept of Predictive Adaptive Responses in humans: biological adaptations to prenatal environmental cues can help prepare for later-life environmental conditions.

Prenatal exposures such as maternal nutrition, air pollution, or heat are well-documented determinants of health across the life course[1–4]. When studying the developmental origins of health and disease, the concept of developmental or phenotypic plasticity is central. It posits that organisms adapt to their environments based on cues encountered in early life stages. Epigenetics plays a major role in this: different phenotypes can be generated from the same genome via the regulation of gene expression, i.e., without changes to the DNA sequence. This regulation of gene expression starts in the prenatal period, when it is thought to be shaped by so-called Predictive

Adaptive Responses (PARs). PARs refer to the mechanisms underlying organisms' adaptations to prenatal environmental signals in ways that optimize preparedness for anticipated postnatal conditions[5,6]. From an evolutionary perspective, PARs enhance species' survival by promoting phenotypes that are better adapted to the anticipated postnatal environment, thereby increasing reproductive success[7,8]. Developmental plasticity enables rapid adaptation to changing environmental conditions and can be advantageous if prenatal signals accurately predict the later-life environment. Conversely, mismatches between prenatal and postnatal conditions

[1]Johannes Gutenberg University Mainz, Chair of Statistics and Econometrics, Mainz, Germany. [2]Heidelberg Medical Faculty and University Hospital, Heidelberg Institute of Global Health, Heidelberg, Germany. [3]Division of Primary Care and Population Health, Stanford University, Stanford, CA, USA. [4]Charité – Universitätsmedizin Berlin, corporate member of Freie Universität Berlin and Humboldt-Universität zu Berlin, Institute of Public Health, Berlin, Germany. [5]Potsdam Institute for Climate Impact Research, Member of the Leibniz Association, Potsdam, Germany. [6]Present address: Institute of Public Health and Nursing Research, Department of Social Epidemiology, University of Bremen, Bremen, Germany. ✉e-mail: pradella@uni-mainz.de

may result in maladaptive outcomes[5,7,9]. However, an adverse exposure may also weaken the physiological system, leaving it more vulnerable to subsequent insults. This mechanism parallels findings from research on later-life adverse conditions, where repeated or prolonged exposure to adverse circumstances, e.g., poor sleep and an unhealthy diet, has been shown to exacerbate health effects[10]. At the same time, the potential for beneficial interaction effects with matching prenatal and postnatal conditions highlights the importance of considering both prenatal and postnatal environments when conducting research on prenatal exposures. However, the role of the postnatal environment in co-shaping responses to prenatal exposures has to date received very limited attention in the epidemiological literature.

Evidence from the animal world supports the PAR framework[11–18]. Adaptive phenotypic evolution is, for example, documented in snowshoe hare offspring who exhibit heightened stress responsiveness when exposed to high prenatal predation risk[19], in sea snail populations that can adapt their colors to their coral hosts, which enhances chances of survival after migration[20], or in chickens who show an improved heat tolerance upon exposure to higher egg incubation temperatures[21]. In research on humans, the focus has predominantly been on how adverse prenatal conditions (e.g., hunger) lead to adverse long-term health outcomes in postnatal environments that were more advantageous than predicted by prenatal circumstances. This is also due to the exposures studied. Since experiments are ethically unfeasible, causal evidence on prenatal exposures requires quasi-random variation in prenatal circumstances in order to separate exposure effects from correlated background characteristics. Researchers have addressed this by utilizing clearly demarcated, unanticipated historical periods of extreme adverse circumstances during pregnancy, such as the Dutch Famine, the Chinese Great Leap Forward, or the Siege of Leningrad[22].

While these studies have made important contributions to understanding the long-term impacts of prenatal adverse events, they only capture a part of the PAR framework. Firstly, research has remained limited to demonstrating damage in response to situations where (adverse) prenatal and (advantageous) postnatal circumstances did not match, rather than testing the full spectrum of the PAR framework's predictions—particularly potential advantages associated with matching prenatal and postnatal circumstances. Secondly, the prenatal exposures studied were often so severe that they may exceed the limits of developmental plasticity, instead causing damage to the developing organism (e.g., extreme caloric restrictions during a famine).

More recently, associations between less extreme prenatal exposures and health over the life course have been documented[3]. These include the adverse impact of droughts during pregnancy on cognitive health in settings where nutrition is dependent on rain-fed agriculture—such as rural India[23]. In drought years, households in rural India own fewer assets and reduce their food intake[24–26]. Crucially, in contrast to extreme conditions such as famines, prenatal drought-related nutritional challenges are more frequently followed by similar postnatal circumstances. Early-life nutrition is critical for cognitive development, alongside genetic factors and child stimulation[27–31]. Cognitive and other neurological functions are particularly shaped by rapid brain growth in prenatal as well as early postnatal life. Brain development starts with the formation of the neural plate around 22 days after conception[32–35] and continues postnatally, with a large share of the brain's ultimate structure and capacity being shaped before age three. Previous studies have shown that droughts during pregnancy and in the first years of life are associated with worse test scores in rural India[36]. Similar observations have also been made in other settings characterized by a high reliance on rainfed agriculture[37–39].

With this study, we seek to address the gap in empirical evidence on PARs in humans by assessing whether prenatal and postnatal droughts interact in shaping cognitive health. Cognitive traits are evolutionarily relevant in the framework of PARs as they influence the likelihood to thrive, to procreate, and to successfully raise offspring[40]. Our study population consists of ~2 million 11-to-16-year-old adolescents assessed in math and reading by the Annual Status of Education Report (ASER) survey in rural areas of India. We use variation in drought occurrences within districts as a proxy for the early-life nutritional environment and leverage this variation to assess whether children were better adapted to postnatal drought experiences if they had already encountered similar conditions prenatally. Since droughts occur quasi-randomly within geographical areas during both the prenatal and the postnatal periods, this allows us to provide empirical causal evidence on the PAR framework in humans.

Our results indicate that early-life drought exposure is associated with adverse cognitive outcomes. However, consistent with the PAR framework, prenatal drought exposure mitigates the negative cognitive consequences of postnatal drought exposure. This finding supports the presence of developmental plasticity in response to environmental conditions in humans and highlights the importance of considering interactions between prenatal and postnatal exposures when assessing the longer-term impacts of early-life conditions.

## Methods
### Dataset
We obtained data from the Annual Status of Education Report (ASER) from 2007 to 2018. ASER is a repeated cross-sectional household-based cognitive assessment conducted across rural India, organized by the Pratham Education Foundation[41]. It was fielded annually from 2007 to 2014, and biannually since. ASER provides data on the math and reading skills of children who are tested orally by trained enumerators[42]. All children are confronted with the same standard test, independent of their age and school enrolment status. They are assessed in the language of their choice, with almost 20 languages available. Rural areas are defined based on the Indian Census 2011 village directory, which identifies rural areas in all Indian districts and serves as sampling frame. ASER data are representative at district, state, and national levels. Our study population consisted of all 11–16-year-old children who were assessed as part of ASER between 2007 and 2018 in at least one test domain (reading, math), leading to a sample size of ~2 million observations. The 11–16 age group was chosen because test scores are more stable and exhibit less day-to-day variability in older children. The available number of children with an assessment in reading was slightly higher than the number of children assessed in math because the ASER assessment usually starts with reading, and some children drop out after the first test, e.g., due to other obligations (communication with ASER survey team, available on request).

The ASER data were collected in accordance with ethical research practices, including voluntary participation and informed consent. Institutional review was not required because the ASER survey does not collect medical or clinical data. All persons involved in data collection and Pratham/ASER Centre associates are signatories to the organization's Child Protection Policy (CPP) and adhere to their responsibilities under the Digital Personal Data Protection Act (DPDPA)[43]. Informed consent to participate in the study was obtained from all participants.

### Cognitive function
We used three indicators of test performance: total performance score, math score, and reading score. In both math and reading, children are presented with problems of ascending difficulty and are marked at the highest level they can successfully solve. The grading system ranges from 0 to 4 for each test domain, with 0 indicating that the child was not able to complete the lowest level (recognizing a letter/1-digit number) and 4 indicating that the child reached the highest level (reading a story/multiplication and division). We additionally constructed a total performance score by summing the math and reading scores, which thus ranges from 0 to 8.

Besides the continuous outcomes, we created binary indicators as outcomes, reflecting the probability of children reaching the highest and second-highest score in each domain. The binary cutoffs were defined as $\geq 7$ and 8 (out of 8) points for the total performance score and $\geq 3$ and 4 (out of 4) points for math and reading. The binary outcomes allow us to focus on the upper end of the performance distribution to examine whether drought exposure in early life particularly affects the likelihood of reaching top scores. This was motivated by the relatively high average scores across all

**Table 1 | Characteristics of 2,032,917 children living in rural India (ages 11–16), Annual Status of Education Report (ASER), 2007–2018**

| | Total | | | Girls | | | Boys | | |
|---|---|---|---|---|---|---|---|---|---|
| | Mean | SD | N | Mean | SD | N | Mean | SD | N |
| **Cognition test results** | | | | | | | | | |
| Total Score | 6.63 | 1.80 | 1,980,249 | 6.55 | 1.85 | 926,589 | 6.70 | 1.75 | 1,035,877 |
| Total Score 8/8 Points | 0.47 | | | 0.45 | | | 0.50 | | |
| Total Score ≥ 7/8 Points | 0.65 | | | 0.63 | | | 0.66 | | |
| Math Score | 3.17 | 1.00 | 1,984,990 | 3.10 | 1.03 | 928,751 | 3.22 | 0.97 | 1,038,379 |
| Math Score: 4/4 Points | 0.50 | | | 0.48 | | | 0.53 | | |
| Math Score ≥ 3/4 Points | 0.75 | | | 0.72 | | | 0.77 | | |
| Reading Score | 3.41 | 1.06 | 2,028,176 | 3.39 | 1.09 | 951,059 | 3.43 | 1.03 | 1,058,774 |
| Reading Score: 4/4 Points | 0.69 | | | 0.69 | | | 0.69 | | |
| Reading Score ≥ 3/4 Points | 0.84 | | | 0.83 | | | 0.84 | | |
| **Child and household characteristics** | | | | | | | | | |
| Age | 13.31 | 1.63 | 2,032,917 | 13.31 | 1.62 | 953,221 | 13.30 | 1.63 | 1,061,276 |
| Child is a boy | 0.53 | | 2,014,497 | 0 | | 953,221 | 1 | | 1,061,276 |
| Currently enrolled in school | 0.93 | | 2,032,917 | 0.92 | | 953,221 | 0.94 | | 1,061,276 |
| • Public school | 0.66 | | | 0.68 | | | 0.65 | | |
| • Private school | 0.26 | | | 0.24 | | | 0.28 | | |
| • Other | 0.01 | | | 0.01 | | | 0.01 | | |
| Never enrolled in school | 0.01 | | 2,032,917 | 0.01 | | 953,221 | 0.01 | | 1,061,276 |
| Mother attended school | 0.48 | | 1,969,196 | 0.48 | | 923,113 | 0.47 | | 1,029,275 |
| Household has electricity | 0.76 | | 1,770,660 | 0.77 | | 834,605 | 0.76 | | 917,916 |

Variables for which no SD is shown are binary variables with means referring to proportions.

outcome measures and remaining variation at the top of the distribution—for example, only 65% of children reached at least 7 out of 8 points on the total score, and only 47% achieved 8 out of 8 points (Table 1).

**Drought exposure**

Drought exposure in utero, as well as for each of the first three years of life, was assigned to each child based on their estimated birth year and district of assessment. The focus on exposure during the first three years of life was motivated by two reasons. Firstly, the prenatal period and the first three years of life are most important for brain formation[25,26]. Secondly, in the context of rural India, droughts in later childhood can also represent a schooling opportunity (in addition to being a nutritional challenge) since during droughts, employment opportunities (such as in agriculture) are reduced. Children are therefore more likely to regularly attend school in drought years, with positive effects on cognition assessment scores[36,44]. This starts to play a role quite early on in this setting, since children are usually not formally employed, but e.g., sent to work at other households at ages as young as 5 years[45,46].

Following the literature, we estimated each child's birth year by subtracting their age at assessment from the interview year, since the ASER dataset does not include birth dates[36]. This is the most precise approach for assigning birth dates, since the ASER survey is conducted at the end of the year (between September and November) in all states every year. We defined in utero exposure as the year prior to the birth year (birth year −1), and postnatal exposures in the first, second, and third years of life as birth year +1, +2, and +3, respectively.

The finest geographical level of survey participants' locations reported in ASER is the district. To define drought years for each district, we used historical gridded rainfall data from the University of Delaware (Terrestrial Precipitation Gridded Monthly Timeseries) as provided by the NOAA Physical Sciences Laboratory (PSL)[47]. We summed monthly measures into yearly rainfall measures. We then calculated rainfall levels for each district

and year using QGIS software, based on a shapefile depicting India's 2001 district boundaries obtained from *ML Infomap*[48], which align with the ASER survey's sampling frame. Specifically, we calculated district-level exposure based on all pixels within a district using Inverse-Distance Weighted (IDW) Interpolation. Thereby, weights are proportional to the inverse of the distance between the district center and the rainfall point. In India, districts have been split due to administrative reasons, with the number of districts increasing from 593 in 2001 to 693 in 2021. To accurately assign exposure, we followed district divisions backward and assigned each child to the district according to the Census 2001, which has remained the ASER sampling frame up until today.

We then calculated droughts based on deviations from the district-specific long-term rainfall mean. Following the literature, we define all years in which rainfall levels were below the 20th percentile of the district-specific long-term mean (1975–2017) to be drought years, a meteorological definition of drought[36,49]. Droughts in India are associated with crop failure and reduced agricultural income, as well as a less nutritious diet among rural households[24–26]. While household assets and nutrition have been shown to be affected by current-year droughts[24–26,37,50], there is no evidence for selective fertility or migration in response to droughts[25,51]. This implies that droughts in rural India can be thought of as quasi-random occurrences.

The definition of drought occurrences based on deviations from district-specific long-term means is pivotal for estimating causal effects. Estimating the causal effect of prenatal environmental exposures is challenging, since the effect of the exposure needs to be disentangled from correlated background characteristics. For instance, regions susceptible to droughts often also have lower economic development indicators and worse health outcomes than other regions—irrespective of drought occurrences. By contrast, rainfall within regions varies quasi-randomly over the years. Defining drought occurrences at district level thus allows us to rely on the quasi-randomness of rainfall variation.

## Statistics and reproducibility

To examine how prenatal and early postnatal drought exposure interact in shaping adolescent cognitive health outcomes, we estimate the following OLS regression using Stata 17.

$$Score_{idty} = \alpha_d + \beta_1 DroughtPrenatal_{idt} + \beta_2 DroughtBirthyear_{idt} +$$
$$\Sigma_{j=1}^3 \beta_j Drought123_{idtj} + \Sigma_{k=1}^3 \beta_k DroughtPrenatal_{idt} * Drought123_{idtk} +$$
$$X'_{ity}\, \gamma + v_{idty}$$

where $Score_{idty}$ is the assessment score of child $i$, assessed in district $d$, born in year $t$ and assessed in year $y$. The $\alpha$ are district fixed effects. By including district fixed effects in our models, we only compare cognitive test score outcomes of adolescents living in the same district (but assessed in different years), further ensuring that only quasi-random rainfall variation identifies any effects. District fixed effects are important to dealing with potential confounding due to district characteristics being correlated with drought occurrences. However, including fixed effects in a logistic regression is technically problematic, so we used OLS in a linear probability model for our binary outcomes. Coefficients from these models should be interpreted as percental differences.

$DroughtPrenatal$ is a binary variable describing whether a drought was experienced in utero. $DroughtBirthyear$ refers to drought exposure in the estimated birth year, while $Drought123$ is a set of dummies indicating drought shocks in the first three years after the birth year. These are the so-called main effects in our model, which are, however, not of our main interest in this study. The $\beta_k$ are the coefficients of the interaction between prenatal and early-life postnatal drought exposures – the coefficients of interest in this study. $X$ is a set of controls, which includes the child's age at assessment, child sex, and a time trend (year of assessment and year of assessment squared), and $v$ is the error term.

The coefficients of interest in the statistical model are the interaction terms $\beta_k$ between prenatal drought exposure and drought exposure in the first postnatal life years, which test whether children faced with a drought in early postnatal life are better prepared to counteract the associated postnatal nutritional shock if they had already experienced a drought while in utero. We interacted prenatal drought exposure with exposures in the first, second, and third years after birth year. As the ASER dataset does not provide birth dates, and the estimated birth year partially overlaps with the prenatal period, we did not include an interaction term with birth year exposure to avoid conflating prenatal and postnatal exposures. This ensures that the interaction terms exclusively capture the interactions between prenatal and postnatal drought exposures, while still accounting for exposure in all early-life years, including the birth year. In heterogeneity analyses, we also present results stratified by child sex.

Since exposure was measured as an occurrence of an overlap between a quasi-randomly occurring drought in pregnancy (defined based on within-district rainfall variation) as well as the early life years, this study can be regarded as a quasi-experiment. The quasi-random exposure to droughts means that potential confounders such as maternal characteristics are canceled out; that is, even though we cannot control for all possible factors that might influence test score outcomes, this does not bias our results, since these factors do not affect whether a drought occurred at a particular life stage. While early-life drought exposures cannot be randomized for ethical reasons, overlap between drought in early life occurs quasi-randomly in a setting with low migration patterns.

To ascertain that exposure status is indeed not correlated with potential confounders, we conducted a negative control test using maternal characteristics reported by ASER. There was no evidence for associations between drought occurrences and maternal characteristics (Table SI 1, Online Appendix).

While droughts are generally associated with nutritional challenges in rural India, this does not necessarily apply to all households. For this reason, our estimation is an intention-to-treat analysis, in which all children whose early life years overlapped with a drought in their district were classified as exposed at that point in time.

We tested the stability of the results by replacing the continuous time trend by dummies for each assessment year (year of assessment fixed effects). We also present the results of analyses in which we adjusted for heavy rainfall in utero and the first years of life (above the 95th district-specific percentile), since heavy rainfall can have detrimental effects on nutrition via e.g., crop damage, soil erosion, or damage to the transport infrastructure due to flooding. Moreover, since there is evidence suggesting that food is sometimes allocated with priority to firstborns in rural India[52,53], we show regressions in which we adjusted for firstborn status.

## Results

Characteristics of the children in our sample and the households in which they reside are summarized in Table 1. The mean age of included children at the time of cognitive assessment was 13.3 years. On average, children achieved 6.63 out of 8 points (total performance). The mean scores achieved in math and reading were 3.17 and 3.41, respectively.

### Associations between early-life drought exposures and cognitive function

Figure 1 displays the main results. Panel A displays effects on continuous outcome measures, while Panels B and C show effects of early-life drought experience on the probability of reaching the full score (Panel B) or missing maximum one point in the respective outcome categories (Panel C). The comparison group in all analyses comprises children who neither experienced a drought prenatally nor in the first 3 years of life.

The main effects of prenatal exposure to droughts and test scores in adolescence are negative across all outcome measures. Children who experienced a prenatal drought reach fewer points in total performance (−0.04, 95% CI: −0.07, −0.01), reading (−0.03, 95% CI: −0.05, −0.01), and math scores (−0.02, 95% CI: −0.04, −0.001) than those who did not. Similarly, for the binary outcome measures (Panels B and C), main effects of early-life drought exposure are in a negative direction for both the prenatal and early childhood periods, most consistently for the total score and math, with evidence being not as strong as for the linear outcomes.

### Interactions between prenatal and postnatal drought exposures

The coefficients of interest in this study are the interaction terms, which test whether prenatal and postnatal drought exposures interact in shaping cognitive performance in adolescence (1st Year of Life x In Utero, 2nd Year of Life x In Utero, 3rd Year of Life x In Utero). Negative interaction terms would indicate that childhood exposures have a stronger effect among those who had already experienced a drought prenatally. In contrast, positive interaction terms suggest that early childhood drought exposures have less negative effects if a drought had already been experienced prenatally, thus implying that matching prenatal and postnatal conditions mitigates the impacts of drought exposures in the later period.

We find consistently positive interaction terms, suggesting that although prenatal droughts impair cognitive development, they reduce potential adverse effects of further drought exposures in childhood. For example, among children who were not exposed to a drought in utero, those exposed to a drought in their second year of life have a 1.44% lower probability of reaching the full total score than those not exposed then (95% CI: −2.20%, −0.68%) (Fig. 1, Panel B). This adverse effect is alleviated among children who, prior to this postnatal exposure, had already experienced a drought prenatally—as indicated by the positive interaction term (2.80%, 95% CI: 0.89%, 4.71%). While being most pronounced for the binary outcome measures (Fig. 1, Panels B and C), this pattern is similar for continuous outcomes (Fig. 1, Panel A). The effect pattern is consistent across reading and math scores, and it is clearest for total score and math performance (Fig. 1). Detailed regression outputs are available in the online appendix (Table SI 2 in the Online Appendix).

Analyses stratified by child sex show similar patterns among girls and boys (Fig. 2).

**Fig. 1 | Prenatal and early-life drought exposures: main effects and interaction effects on cognitive function among 11-16-year-old adolescents in rural India.** Forest plots displaying the effects of prenatal and early-life drought exposures on cognitive function (**A**: impacts on points reached in the respective outcome category, **B**, **C**: change in the probability of reaching the indicated number of points in the respective outcome category). Within each panel, Total Score, Reading, and Math represent results from separate regression models, each corresponding to the outcome category indicated in the panel title. The coefficients of interest are the interaction terms between prenatal and early-life exposure (In Utero × 1st, 2nd, and 3rd Year of Life). All estimates are adjusted for child sex, age at assessment, and a time trend (year of assessment and its square). Effects were estimated using multivariable regression models (OLS) with two-sided hypothesis tests; the error bars in the figures depict 95 percent confidence levels.

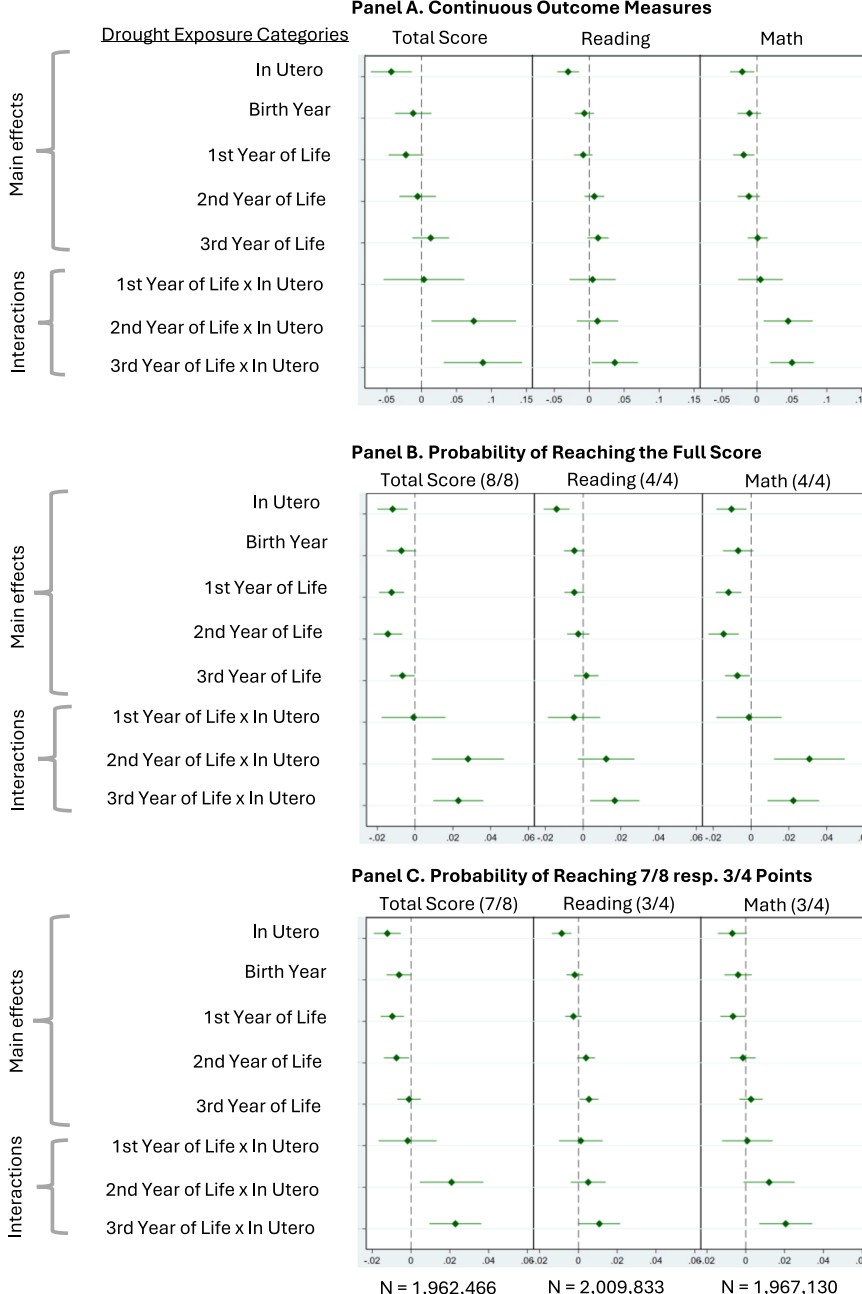

## Sensitivity
The patterns of associations were stable to replacing the time trend with year of assessment dummies (Fig. SI 1, Online Appendix), additionally adjusting for heavy rainfall at all ages of exposure (Fig. SI 2, Online Appendix) and adjusting for being a firstborn child (Fig. SI 3, Online Appendix).

## Discussion
This study is, to our knowledge, the first to systematically assess prenatal-postnatal interactions of similar-type exposures in humans within the Predictive Adaptive Responses framework. Consistent with the previous literature, we found that early-life drought exposure negatively impacted cognitive function[36–39]. In line with our hypothesis, there was a positive interaction between prenatal drought exposure and postnatal drought exposure (in the second and third years of life) on cognitive function. This suggests that children who experienced a drought in early childhood were better prepared to cope with the

drought-related nutritional scarcity if they had already experienced similar circumstances prenatally.

Given advances in research on the Developmental Origins of Health and Disease (DOHaD) over the past two decades, effects of less extreme prenatal exposures on later-life outcomes are of increasing interest. It has been shown that income shocks, nutritional supplementation, stress, or weather fluctuations (e.g., hotter temperatures) are associated with a diverse range of outcomes, including school performance and anthropometric measures in childhood and adulthood[3]. Such prenatal circumstances are more likely to be followed by similar circumstances later in life than extreme exposures such as famines studied by much of the earlier DOHaD literature[22]. A deeper understanding of phenotypic plasticity and adaptive capacities is also essential for advancing our understanding of population-level adaptations to environmental change.

Our findings indicate some degree of adaptive capacity to changing environmental circumstances. While suboptimal nutrition during the

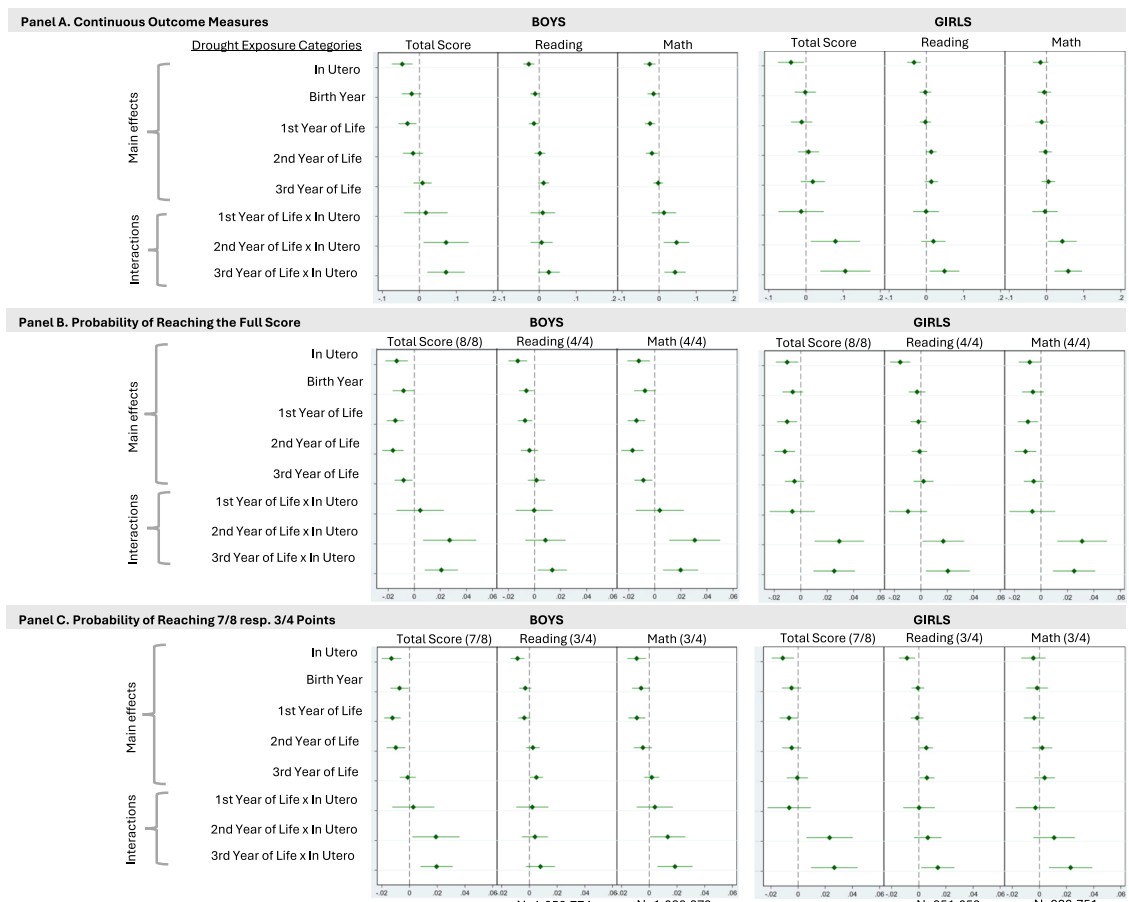

**Fig. 2 | Prenatal and early-life drought exposures: main effects and interaction effects on cognitive function among 11–16-year-old adolescents in rural India, stratified by child sex.** Forest plots displaying the effects of prenatal and early-life drought exposures on cognitive function, stratified by child sex (**A**: impacts on points reached in the respective outcome category, **B**, **C**: change in the probability of reaching the indicated number of points in the respective outcome category). Within each panel, Total Score, Reading, and Math represent results from separate regression models, each corresponding to the outcome category indicated in the panel title. The coefficients of interest are the interaction terms between prenatal and early-life exposure (In Utero × 1st, 2nd, or 3rd Year of Life). All estimates are adjusted for age at assessment and a time trend (year of assessment and its square). Effects were estimated using multivariable regression models (OLS) with two-sided hypothesis tests; the error bars in the figures depict 95 percent confidence levels.

prenatal phase can hamper cognitive development, at the same time, this prenatal adversity can to some degree protect against similar nutritional adversity postnatally—which has been described as Predictive Adaptive Response (PAR)[6,7]. Drought-related nutritional challenges can be comparatively mild prenatal shocks, making it more likely that adaptions are possible and prenatal exposure does not merely damage the organism[7]. This notion is supported by literature on epigenetic alterations in response to changing nutritional circumstances in early life. Several studies have demonstrated that maternal nutrition during pregnancy and breastfeeding can affect neurological development through a range of epigenetic mechanisms[54,55]. Recent research also indicates that maternal diet during pregnancy can have effects on child neurodevelopmental outcomes as early as at 40 days of age[56]. While this suggests that there is a direct link between early-life nutrition and later cognitive skills, indirect links are possible, too. For example, poor prenatal nutrition may influence characteristics such as persistence or inquisitiveness, which in turn affects children's cognitive development. To our knowledge, this indirect pathway has not been investigated yet and warrants future research. Our study examined the effects of similar prenatal and postnatal exposures without being able to disentangle the specific underlying mechanisms. Future work will need to further elucidate the exact dynamics e.g., whether it is caloric deficiencies that matter, changed intakes of certain micronutrients during drought periods, or other pathways.

Research on interactions between prenatal and postnatal exposures in humans has only recently begun to emerge. Few studies take postnatal circumstances into consideration when assessing the impacts of prenatal exposures. Some studies have investigated early-life exposures with less potential for biological adaptations that can prepare for similar circumstances. For example, air pollution more likely leads to direct fetal damage rather than phenotypic adaptation for future air pollution tolerance[57]. Other studies investigated interactions between exposures along different dimensions. For example, in Indonesia, prenatal nutritional stress led to respiratory disease primarily among persons who smoke and to lower height in those faced with unimproved sanitary conditions—i.e., the moderating exposures are along different dimensions so that a PAR is not expected[58,59]. Finally, a methodological limitation of most previous studies is that the postnatal exposure was not (quasi-)random, thus limiting causal inference. In contrast, in this study, we specifically examined exposures within the same domain during both prenatal and early postnatal life, and adopted a quasi-experimental framework to allow for causal interpretation of our results.

Future studies of PARs in humans will thus need to carefully distinguish between exposures that may initiate developmental plasticity, i.e., provide room for beneficial biological adaptations in a changed postnatal environment, and exposures that may be solely damaging. Thereby, systematically assessing PARs in humans requires context sensitivity related to

exposures and outcomes. For example, agriculture in India is highly rainfall-dependent and relevant for nutrition. Diets in rural India are often largely based on cereals and exhibit low diversity[60]. Drought-induced agricultural shortfalls may further decrease dietary diversity and exacerbate micronutrient deficiencies, and micronutrients are critical for children's cognitive development[61–63]. Drought-related nutritional challenges have also been documented in various other low-income contexts around the globe[64–69]. However, it is doubtful if similar dynamics can be expected in settings with higher baseline food security. Moreover, matching vs mismatching pre- and postnatal exposures can also occur with respect to other types of exposures. The literature on this has remained scarce. One of the few available studies is a cohort study on 221 pregnant women in the United States focusing on exposure to maternal depression[70]. This study—like ours—suggests that matching pre- and postnatal circumstances is beneficial for health: infants had poorer cognitive skills both when maternal depression occurred either prenatally or postnatally. However, when the mother reported depressive symptoms during both periods (i.e., a matching exposure for her child), infant outcomes were comparable to those of infants whose mother never experienced a depression[70].

The interaction between prenatal and early postnatal nutritional circumstances in our study is only apparent in the second and third year of life, but not the first. A potential explanation could be breastfeeding, which is common in rural India, also beyond the first 6 months, and provides a range of micronutrients, so that the impacts of food scarcity might matter less during this young age[71]. We did not study interactions between prenatal and postnatal exposures beyond the first three years of life. Future research is encouraged to examine if cognitive function responds to nutritional impacts also at later life stages. In our case, this was not feasible, since drought experiences in rural India are associated with a higher probability of regularly attending school among older children—which would bias our results. Additionally, as previous research has shown that poor nutrition during pregnancy has a range of impacts, from child growth to a broad set of adult health outcomes[22], it will be interesting to study interaction effects for other outcomes besides cognitive development. Future research is also encouraged to examine interactions between early-life exposures that are not expected to trigger developmental plasticity (e.g., prenatal nutrition and postnatal air pollution or other clearly different exposures) to further test the predictions of the PARs framework.

A few limitations to our study need to be noted. First, some households are not dependent on agricultural produce or income and thus are not much affected by droughts, which leads to an underestimation of the true effect size in our intention-to-treat framework. Furthermore, the finest geographical level available in the ASER dataset was the district, which means that household drought exposure was only approximated, again leading to non-differential misclassification and a dilution of effect sizes. Second, we studied impacts, but cannot speak to the precise pathways leading to these. Droughts have primarily been linked to food availability and nutritional intake in rural India; however, they have also being associated with limited access to safe drinking water and increased risk for infectious diseases, which may be other potential pathways. Our theoretical background lies in the PAR framework, which centers on epigenetic adaptations to environmental circumstances. However, we could not test any epigenetic effects directly, so that the exact mechanisms in essence remain a biological "black box". It is possible that developmental plasticity involves other adaptive processes besides epigenetics. Third, while ASER uses standardized tests for the assessment of math and reading skills, more sophisticated assessments—such as cognitive assessment scales, memory process tests, or visual perceptual skill tests—are not feasible at this scale, and could thus not be studied.

## Summary

During early life, organisms react to environmental cues via biological adaptations. This phenotypic plasticity implies that prenatal conditions can shape to what extent adverse experiences in postnatal life affect health. Our findings suggest that adverse postnatal environments may indeed less strongly affect health if they were preceded by similar prenatal conditions. The relevance of our findings extends beyond medical theory. A substantial share of the global population relies on rainfed agriculture and is directly impacted by rainfall fluctuations. As the climate crisis progresses, the variability of weather patterns as well as climate-related hazards is predicted to increase[72]. While our study suggests that similar prenatal and postnatal conditions may offer some benefits through epigenetic adaptations, such alignment cannot be planned or anticipated. Children will more often experience mismatches in prenatal and postnatal conditions, and additionally, the capacity of adaptive responses to mitigate adverse effects may be limited under more extreme climatological anomalies.

## Data availability

The data supporting the findings of this study are available from the ASER Centre (asercentre.org), the measurement, assessment, and research unit of the Pratham Education Foundation. Restrictions apply to the availability of the data, and they are therefore not publicly accessible. Researchers interested in obtaining data access for research purposes may contact the ASER Centre via email. The source data for the main figures (Figs. 1 and 2) is available in the Supplementary Data file. As indicated in the Supplementary Data file, Sheet 1 corresponds to Fig. 1, while Sheets 2 and 3 correspond to Fig. 2.

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

## Acknowledgements
This work was supported by the German Research Foundation (project number 455841434). F.P. and R.v.E. were supported by the High-Potential Research Areas "EXPOHEALTH" and "Interdisciplinary Public Policy" at the Johannes Gutenberg University Mainz, which are funded by the Ministry of Science and Health of Rhineland-Palatinate, Germany. We thank Tabea Schubert for excellent research assistance in spatial data management.

## Author contributions
All authors contributed to the study conception and design. Data curation, coding, and analysis were performed by F.P., with input from R.v.E. All authors engaged in the interpretation of results. The first draft of the manuscript was written by F.P., and all authors commented on previous versions of the manuscript. All authors read and approved the final manuscript.

## Funding

## Competing interests
The authors declare no competing interests.
