## [Transparent Peer Review file · Communications Medicine]

Prenatal and postnatal droughts interact in shaping cognitive development

Corresponding Author: Dr Fabienne Pradella

Version 0:

Reviewer comments:

Reviewer #1

(Remarks to the Author)

Summary

This study aimed to test the match mismatch hypothesis which posits that organisms exhibit capacity to alter prenatal physiological development in ways that aim to increase fitness in postnatal environments. When prenatal conditions match postnatal conditions, the organism is likely better equipped to survive to sexual maturity within these postnatal conditions. However, when mismatches occur, the organism is at risk for adverse outcomes. The authors used large population-based data to identify when certain districts in rural India experienced drought conditions, and compared exposure to these conditions to standardized math, reading and total scores (math+reading) in a large sample of children. To test the mismatch hypothesis, the authors examined interactions between when drought conditions were experienced (pre vs. postnatal). While exposure to drought in utero was associated with poorer outcomes, exposure to drought in utero and during the 2nd and 3rd year of life had a somewhat protective effect on cognitive outcomes.

Overall Impressions

The reviewer found the manuscript well done. Justification for the approach was sound, and the methodology enabled authors to test the mis match hypothesis. Potential issues were addressed in sensitivity analyses. The authors addressed limitations of the work, which are important to note, but I don't believe significantly compromise the findings presented herein.

Specific comments

1. It may be helpful for the authors to frame their arguments for why PAR would occur in the instance of drought exposure, and not lead to adverse outcomes as would be predicted by dual hit, or allostatic load theories. The authors state that droughts are more mild, and likely don't lead to damage like other exposure might (the authors note air pollution). I believe presenting their work in the context of these other theories which have consistently shown that multiple exposures increase disease risk would be beneficial.
2. It might also be helpful for the authors to note that PAR likely evolved as a mechanism to increase chances of survival to sexual maturity (and not to increase traits that are beneficial in today's environments-unless these traits also increase changes of survival to sexual maturity). How would better performance on standardized tests reflect the brain changes influenced by the PAR? Some might say that PAR as a function of drought might lead to more aggressive behaviours which may adversely impact performance on these tests (given that prenatal conditions signal a postnatal environment with limited resources, therefore, if one wants to pass on their genes, they need to ensure they get resources they need to survive). Interested in the authors thoughts on how the PAR would shape a brain better equipped to perform on a task that no one would have to excel at in primitive conditions within which the PAR evolved.
3. The authors state that to reveal a PAR, the pre and postnatal exposures likely need to be very similar. I wonder if rather than adjusting for postnatal rainfall, if the authors could test an interaction between prenatal drought and postnatal severe rainfall (or another postnatal adversity). If worse outcomes are observed in those exposed to these two different adversities, this may provide a further justification of match mis match.
4. Line 294, authors state that IGF2 is associated with prenatal nutrition and memory. It is not clear how this relates to the current work? I understand that mechanisms are unclear, but I believe that using the PAR to hypothesize mechanisms might be helpful. Perhaps children exhibit greater resilience/resourcefulness which translates to better cognitive skills? Maybe they exhibit more persistence or motivation to do things they deem important (which may have been evolutionary adaptive

when predicting postnatal environments with low resources)? I think commenting on these might help guide future research on mechanisms.

Reviewer #2

(Remarks to the Author)

This a fascinating paper testing the Predictive Adaptive Response (PAR) to adversity hypothesis. Despite the limitations in this study of unknown measurement error and precision, the results support the somewhat counter-intuitive outcome that early (in utero) exposure to nutritional adversity prepares the child (improving resilience) for later exposures to adversity. The study focus on cognitive outcomes is surprising in view of the adversity variable chosen to examine was probable nutritional exposure. It would be interesting to know if various factors related to children's growth and development also were influenced by early and later mismatches.

The model tested is 2 (exposed; not exposed prenatal) X 2 (exposed; not exposed postnatal). The interactions terms test the PAR hypothesis. Not clear where the main effects are presented. In the text, it was stated that drought did have effects on cognitive outcomes both from early and later exposure. In the Figure 1 panels, I assume a negative score indicate the deleterious effects of drought exposure and reflects the main effect. I assume it is not a difference score from the comparison groups.

Children experiencing prenatal or postnatal exposure to drought have lower cognitive scores than comparison children. But those exposed early are not as affected by later drought conditions as prenatally unexposed children. Is this a Floor Effect, or evidence for the Law of Initial Values? That is, are they starting at a lower level so the only way they can move is up?

There are other models than PAR, including the Biological Sensitivity to Context and the Stress-Inoculation frameworks. Perhaps the paper could be presented in a broader context, but the review and rationale are both clear and accurate. However, there is a human study with precise predictive and outcome measures that are highly consistent with, and supportive of, the findings in this manuscript. The authors may want to consider if or how the depression exposure in humans in that paper, with the same model, is related to their results. (Psychological Science, 2012, 23 (1), 93-100).

Version 1:

Reviewer comments:

Reviewer #1

(Remarks to the Author)

I have now had the opportunity to thoroughly re-review the manuscript and the author's responses to each comment. I feel that the authors provided very thoughtful responses. I now have a clearer understanding of their theoretical approach. They also provide important direction for future work to further test this hypothesis.

Reviewer #2

(Remarks to the Author)

Very thorough and comprehensive response to suggestions.

Editor

We ask that you address all comments made by the reviewers. The editor also suggests less of a focus on epigenetic mechanisms, as the paper does not directly test epigenetic effects and other biological mechanisms could be underlying the results. Additionally, we want to stress the importance of being transparent and making all limitations clear when revising the manuscript.

Thank you for this important remark. We adapted the first paragraph of the manuscript. While theory and previous research predominantly focus on epigenetics, you are right that this need not be the full story. We therefore broadened the scope of this opening paragraph.

We furthermore changed the first sentence of the abstract, which in its previous version had started with the words “Prenatal epigenetic programming ...”. It now reads (changes marked in italics):

“Developmental plasticity refers to biological adaptations, most often prenatally, to environmental cues. These can help organisms adapt to similar postnatal environments, with health benefits if prenatal and postnatal conditions match.” (p.2)

Throughout the manuscript, we furthermore changed “epigenetic adaptations” to “biological adaptations” a few times and deleted the word “epigenetic” once.

Furthermore, we added the following to the limitations part of the Discussion section:

“Our theoretical background lies in the PAR framework which centers on epigenetic adaptations to environmental circumstances. However, we could not test any epigenetic effects directly, so that the exact mechanisms in essence remain a biological “black box”. It is possible that developmental plasticity involves other adaptive processes besides epigenetics.” (p. 12)

We believe we have now made all limitations to our study clear in our manuscript.

Reviewer #1

Summary

This study aimed to test the match mismatch hypothesis which posits that organisms exhibit capacity to alter prenatal physiological development in ways that aim to increase fitness in postnatal environments. When prenatal conditions match postnatal conditions, the organism is likely better equipped to survive to sexual maturity within these postnatal conditions. However, when mismatches occur, the organism is at risk for adverse outcomes. The authors used large population-based data to identify when certain districts in rural India experienced drought conditions, and compared exposure to these conditions to standardized math, reading and total scores (math+reading) in a large sample of children. To test the mismatch hypothesis, the authors examined interactions between when drought conditions were experienced (pre vs. postnatal). While exposure to drought in utero was associated with poorer outcomes, exposure to drought in utero and during the 2nd and 3rd year of life had a somewhat protective effect on cognitive outcomes.

Overall Impressions

The reviewer found the manuscript well done. Justification for the approach was sound, and the methodology enabled authors to test the mis match hypothesis. Potential issues were addressed in sensitivity analyses. The authors addressed limitations of the work, which are important to note, but I don't believe significantly compromise the findings presented herein.

Thank you for the concise summary of our study. We are grateful for your careful review and constructive feedback as well as for your recognition of the importance of our study. We have incorporated several of your suggestions into the revised manuscript, which we believe has helped clarify key points and improved the overall narrative. A detailed, point-by-point response is provided below.

Specific comments

1. It may be helpful for the authors to frame their arguments for why PAR would occur in the instance of drought exposure, and not lead to adverse outcomes as would be predicted by dual hit, or allostatic load theories. The authors state that droughts are more mild, and likely don't lead to damage like other exposure might (the authors note air pollution). I believe presenting their work in the context of these other theories which have consistently shown that multiple exposures increase disease risk would be beneficial.

We agree that it is an important concept that a spell of adversity may have a stronger impact if the person had been exposed to adverse circumstances before. We realize that we did not yet elaborate much on this and now added an explanation to the first paragraph of our Introduction section.

We would like to point out here that this idea fully fits with the framework around Predictive Adaptive Responses (PARs) which we utilize throughout this paper. Following theories on PARs, if two consecutive exposures are of a similar type, exposure to the first one can lead to adaptation and confer a protective advantage against the second one. If, however, the first and the second exposure are of different types, then no beneficial adaptation occurs. Hence the effect of the second exposure will not be mitigated. And it may even lead to a stronger effect of the second exposure if the body has been weakened by the first exposure.

Dual hit theory and allostatic load theory are similar in their general idea, to our understanding, but used in different contexts, such as predominantly Parkinson's disease for dual hit theory (e.g. Hawkes et al. (2009)) and the joint impact of chronic stress exposure and life events for allostatic load theory (e.g. Guidi et al. (2020)). We have not seen research that used these theories for prenatal exposures. PARs appear the more fitting framework for those.

In the text added in response to your comment, we therefore draw parallels with the research you refer to, and in this context cite Guidi et al. (2020), an important paper on allostatic load theory. However, we do not

discuss the details of the specific theories of dual hit and allostatic load due to the reasons outlined above as well as word limits.

Please see below the new text in the Introduction section, with additions indicated in italics for your convenience:

“Developmental plasticity enables rapid adaptations to changing environmental conditions and can be advantageous if prenatal signals accurately predict the later-life environment. Conversely, mismatches between prenatal and postnatal conditions may result in maladaptive outcomes [5, 7, 9]. However, an adverse exposure may also weaken the physiological system, leaving it more vulnerable to subsequent insults. This mechanism parallels findings from research on later-life adverse conditions, where repeated or prolonged exposure to adverse circumstances, e.g. poor sleep and an unhealthy diet, has been shown to exacerbate health effects [10]. At the same time, the potential for beneficial interaction effects with matching prenatal and postnatal conditions highlights the importance of considering both prenatal and postnatal environments when conducting research on prenatal exposures.” (p.3)

2. It might also be helpful for the authors to note that PAR likely evolved as a mechanism to increase chances of survival to sexual maturity (and not to increase traits that are beneficial in today’s environments-unless these traits also increase changes of survival to sexual maturity). How would better performance on standardized tests reflect the brain changes influenced by the PAR? Some might say that PAR as a function of drought might lead to more aggressive behaviours which may adversely impact performance on these tests (given that prenatal conditions signal a postnatal environment with limited resources, therefore, if one wants to pass on their genes, they need to ensure they get resources they need to survive). Interested in the authors thoughts on how the PAR would shape a brain better equipped to perform on a task that no one would have to excel at in primitive conditions within which the PAR evolved.

Thank you for this interesting perspective. We agree with the point in your first sentence that PARs did not evolve to increase traits that are beneficial in today’s environment, but with a slight twist. PARs did not simply evolve to increase chances of survival to sexual maturity, but to increase chances of passing on one’s genes to subsequent generations, i.e. to develop a “phenotype beneficial for reproduction” (Wells, 2012). This is important, because the test scores we studied may not be overly important in their own right but they are a proxy for traits that are beneficial for the probability of passing on one’s genes.

The mere performance on specific questions is thus indeed not what is interesting here from a research perspective. What counts is the cognitive capabilities. One could see the standardized tests as a proxy for this broader concept. The human species has been equipped with high intelligence for a very long time. And good cognitive performance has always been important for humans to thrive in their environments: This includes both the probability to procreate and successfully raise offspring as well as the general ability to successfully navigate challenging circumstances (while aggression may be one potential strategy in challenging circumstances, it is certainly not the only or the most promising strategy, especially on a population level). In our study, cognitive performance was measured using standardized tests. Millennia ago, measuring cognitive performance would have required different tasks.

We therefore argue that PARs do not shape the brain “to perform on a task that no one would have to excel at in primitive conditions”. Rather, the brain may be better able to handle unforeseen and difficult circumstances (which is part of some definitions of intelligence) which in turn increases the chances of survival. Beyond the probability of survival until reproductive age, insults that harm cognitive performance can also reduce the probability to pass on one’s genes since e.g. higher cognitive abilities are associated with higher socioeconomic status, which in turn is linked to a greater likelihood of successfully raising offspring. PARs suggest that in response to prenatal exposures, the body adapts in such a way that it is better prepared for similar exposures in later life. In our setting, exposure to nutritional shortage may “program” the brain to function adequately with limited nutritional resources, so that the damage to cognitive performance from a second such insult will be lessened. This is beneficial in an evolutionary sense for the reasons outlined above.

Your comment made us realize that it is important to elaborate on this point in the manuscript to provide a clear narrative, and to emphasize that the standardized tests employed in ASER are not so much relevant in their own right, but rather for the broader concept of cognitive performance which they intend to measure. We therefore added the following paragraphs to the manuscript.

First, we added to the explanation of the concept of PARs in the introduction:

“PARs refer to the mechanisms underlying organisms’ adaptations to prenatal environmental signals in ways that optimize preparedness for anticipated postnatal conditions [5, 6]. *From an evolutionary perspective, PARs enhance species’ survival by promoting phenotypes that are better adapted to the anticipated postnatal environment, thereby increasing reproductive success [7, 8].*” (p. 3)

We then revisit this issue towards the end of the introduction to explain that cognitive health is part of a phenotype that is beneficial for reproduction to make this clear from the beginning of the manuscript:

“With this study, we seek to address the gap in empirical evidence on PARs in humans by assessing whether prenatal and postnatal drought interact in shaping cognitive health. *Cognitive traits are evolutionarily relevant in the framework of PARs as they influence the likelihood to thrive, to procreate and to successfully raise offspring [40].*”

We decided not to pick up your point about aggression in the manuscript. While you may be right about this, we did not find sufficient backup in the scientific literature and we wanted to refrain from speculating too much in our paper (we identified some literature on associations between current nutrition and aggression; however, the literature does not suggest that early-life nutrition is associated with aggression later in life). While being programmed for life in an environment with scarce nutritional sources may involve being programmed for more aggressive behavior, the opposite may also occur. Several studies have shown that a lack of nutrition in the earliest life phases leads to a reduced height (Karimi et al., 2021; Lassi et al., 2022), suggesting that exposed individuals will be physically weaker. If one is shorter or weaker, being more aggressive may not be beneficial at all. By contrast, one might end up with even less food if trying to cope with scarcity by fighting for food. Cognitive skills, however, may be useful in any difficult situation.

3. The authors state that to reveal a PAR, the pre and postnatal exposures likely need to be very similar. I wonder if rather than adjusting for postnatal rainfall, if the authors could test an interaction between prenatal drought and postnatal severe rainfall (or another postnatal adversity). If worse outcomes are observed in those exposed to these two different adversities, this may provide a further justification of match mis match.

Thank you for this thoughtful suggestion. As we also noted in our response to your first comment, it is indeed the case that the PAR framework predicts that interactions between dissimilar exposures do not produce the beneficial adaptations. As you also state, this highlights the importance of similarity between exposures in eliciting adaptive responses. It is an interesting idea to test responses to different adversities to provide further justification for the PARs framework.

However, the difficulty with testing interaction effects between prenatal droughts and postnatal severe rainfall is that it is not altogether clear that these lead to different adversities. As we also mention in the manuscript (p.7) there are some differences between both types of events, particularly, heavy rainfall can lead to more destruction of buildings and (transport) infrastructure. At the same time, both events are also similar in that they can lead to harvest loss and therefore negatively impact nutrition, even though the mechanisms leading to this may be different (in the case of severe rainfall rather crop damage and soil erosion).

We did run the analysis you suggested and you find the results below (Figure R1). As shown below, we do not observe a consistent pattern of interaction effects between prenatal drought and early-life severe

rainfall exposure. The few statistically significant interaction effects that we observe point to a potential adverse effect of dual exposure, but they might also be due to chance and do not appear consistently. There are several possible explanations for this. As you hypothesized, one reason may be that these exposures represent distinct types of adversity that do not profit from PARs to the prenatal exposure. The idea that those are indeed different exposures is tentatively supported by the main effects which indicate that severe rainfall is positively associated with later cognitive test performance (this is an interesting observation in itself which we might look at in follow-up research but goes beyond the scope of this paper – so thank you again for this suggestion). At the same time, the literature shows that severe rainfall is associated with negative impacts on nutrition, so that it remains unclear what hypothesis we are testing here.

While we find your suggestion highly valuable, we unfortunately do not have access to a sufficiently different type of exposure in our dataset that would allow for a more conclusive test of this hypothesis. Nonetheless, we fully agree that this line of inquiry warrants further exploration in future research. In particular, testing interactions between more clearly distinct exposures such as, for example, prenatal drought and postnatal air pollution, could be an effective way to probe the boundaries of the PARs framework. Future research will need to use more recent data sources for this: high-quality air pollution data have only become widely available in India from the 2010s onwards. Even though some reanalysis efforts by now also cover the mid-2000s (Kawano et al., 2025), we are unable to investigate this using the ASER data (the children we look at were born between 1991 and 2007). Due to the uncertainty surrounding the interpretation of the interaction between drought and severe rainfall, we have opted not to include this additional analysis in the manuscript. However, we agree that your suggestions warrant further attention and have therefore incorporated them into the future research section, as follows:

“Future research is also encouraged to examine interactions between early-life exposures that are not expected to trigger developmental plasticity (e.g., prenatal nutrition and postnatal air pollution or other clearly different exposures) to further test the predictions of the PARs framework.” (p. 11)

Figure R1. Prenatal drought and early postnatal severe rainfall exposures: main effects and interaction effects on cognitive function among 11-16-year-old adolescents in rural India. Forest plot displaying the effects of prenatal drought and early-life severe rainfall exposures on cognitive function (Panel A: impacts on points reached in the respective outcome category, Panels B and C: change in the probability of reaching the indicated number of points in the respective outcome category). Within each panel, Total Score, Reading, and Math represent results from separate regression models, each corresponding to the outcome category indicated in the panel title. The coefficients of interest are the interaction terms between prenatal and early-life exposure (In Utero × 1st, 2nd, and 3rd Year of Life). All estimates are adjusted for child sex, age at assessment and time trend (year of assessment and its square).

4. Line 294, authors state that IGF2 is associated with prenatal nutrition and memory. It is not clear how this relates to the current work? I understand that mechanisms are unclear, but I believe that using the PAR to hypothesize mechanisms might be helpful. Perhaps children exhibit greater resilience/ resourcefulness which translates to better cognitive skills? Maybe they exhibit more persistence or motivation to do things they deem important (which may have been evolutionary adaptive when predicting postnatal environments with low resources)? I think commenting on these might help guide future research on mechanisms.

Thank you for your comment. Upon re-reading our paragraph, we realized that we had indeed not been very clear and that, as you mention, the discussion on IGF2 does not fit the general line of argumentation around PARs in our paper. Furthermore, we had not yet thought of the alternative pathways you describe. We rewrote this paragraph and it now reads as follows (changes in italics):

“Our findings indicate some degree of adaptive capacity to changing environmental circumstances. While suboptimal nutrition during the prenatal phase can hamper cognitive development, at the same time, this prenatal adversity can to some degree protect against similar nutritional adversity postnatally – which has been described as Predictive Adaptive Response (PAR) [6, 7]. Drought-related nutritional challenges *can be* comparatively mild prenatal shocks, making it more likely that adaptations are possible and prenatal exposure does not merely damage the organism [7]. This notion is supported by literature on epigenetic alterations in response to changing nutritional circumstances in early life. *Several studies have demonstrated that maternal nutrition during pregnancy and breastfeeding can affect neurological development through a range of epigenetic mechanisms [52, 53]. Recent research also indicates that maternal diet during pregnancy can have effects on child neurodevelopmental outcomes as early as at 40 days of age [54]. While this suggests that there is a direct link between early-life nutrition and later cognitive skills, indirect links are possible, too. For example, poor prenatal nutrition may influence characteristics such as persistence or inquisitiveness, which in turn affects children’s cognitive development. To our knowledge, this indirect pathway has not been investigated yet and warrants future research.* Our study examined the effects of similar prenatal and postnatal exposures without being able to disentangle the specific underlying mechanisms. *Future work will need to further elucidate the exact dynamics e.g. whether it is caloric deficiencies that matter, changed intakes of certain micronutrients during drought periods, or other pathways.*” (p. 10)

Your comment highlights a broader issue: there are several pathways through which early-life nutrition may influence cognitive function later in life. We agree that this remains understudied. While we are limited in examining these mechanisms in this paper due to data constraints, we now acknowledge both the lack of research on indirect as well as direct pathways in the revised paragraph above.

Reviewer #2

This a fascinating paper testing the Predictive Adaptive Response (PAR) to adversity hypothesis. Despite the limitations in this study of unknown measurement error and precision, the results support the somewhat counter-intuitive outcome that early (in utero) exposure to nutritional adversity prepares the child (improving resilience) for later exposures to adversity.

Thank you for your appreciation of our research. We are pleased to hear that you found the study fascinating and are grateful for the constructive feedback you provided. Please find our detailed, point-by-point responses to your comments below.

1. The study focus on cognitive outcomes is surprising in view of the adversity variable chosen to examine was probable nutritional exposure. It would be interesting to know if various factors related to children's growth and development also were influenced by early and later mismatches.

Thank you for highlighting that we have not been clear enough on this point. Please note that the choice of cognitive outcomes is not uncommon in studies on effects of early-life nutritional exposures. One reason is that cognitive development is generally considered one of the indicators for child development (World Health Organization, 2023). In fact, cognitive development is of major interest in this context as part of the "stunting syndrome" for which short stature can be considered a (more quickly measured) proxy (De Onis & Branca, 2016; Prendergast & Humphrey, 2014).

In the Introduction section, we relate to the previous literature as follows:

“Early-life nutrition is critical for cognitive development, alongside genetic factors and child stimulation [27-31]. Cognitive and other neurological functions are particularly shaped by rapid brain growth in prenatal as well as early postnatal life. Brain development starts with the formation of the neural plate around 22 days after conception [32-35] and continues postnatally, with a large share of the brain's ultimate structure and capacity being shaped before age three. Previous studies have shown that droughts during pregnancy and in the first years of life are associated with worse test scores in rural India [36]. Similar observations have also been made in other settings characterized by a high reliance on rainfed agriculture [37-39].” (p. 4)

To make it clearer that there are known biological mechanisms leading from early-life nutrition to cognitive outcomes, we now added the following sentences:

“Several studies have demonstrated that maternal nutrition during pregnancy and breastfeeding can affect neurological development through a range of epigenetic mechanisms [52, 53]. Recent research also indicates that maternal diet during pregnancy can have effects on child neurodevelopmental outcomes as early as at 40 days of age [54]. While this suggests that there is a direct link between early-life nutrition and later cognitive skills, indirect links are possible, too. For example, poor prenatal nutrition may influence characteristics such as persistence or inquisitiveness, which in turn affects children's cognitive development. To our knowledge, this indirect pathway has not been investigated yet and warrants future research.” (p. 10)

We agree with you that it would be highly interesting to assess whether there are prenatal-postnatal interactions with respect to growth and other child developmental outcomes besides cognition. For example, it is well-known that poor prenatal nutritional circumstances can reduce height growth (e.g. Lumey et al. (2011)). However, unfortunately, our dataset does not contain information on children's height, nor does it contain information on other developmental outcomes. This is a known issue with the ASER data and related to the study design: The ASER survey was designed as a rapid assessment with standardized tests that can be administered by volunteers in the rural communities in a cost efficient way (Banerji et al., 2013). Consequently, no measurements beyond the ones specified in the manuscript are available. At the same time, the unavailability of the relevant outcome information in our dataset of course does not make the point you raised less relevant and we realized that we had not been clear enough in the manuscript. We agree that future research should be encouraged to use other large datasets which contain information on other outcomes (e.g. child height) to study whether prenatal-postnatal interaction is present also for those

outcomes. We now more specifically point this out in the suggested directions for future research in the Discussion section:

“Additionally, as previous research has shown that poor nutrition during pregnancy has a range of impacts from child growth to a broad set of adult health outcomes [22], it will be interesting to study interaction effects for other outcomes besides cognitive development.” (p. 11)

2. The model tested is 2 (exposed; not exposed prenatal) X 2 (exposed; not exposed postnatal). The interactions terms test the PAR hypothesis. Not clear where the main effects are presented. In the text, it was stated that drought did have effects on cognitive outcomes both from early and later exposure. In the Figure 1 panels, I assume a negative score indicate the deleterious effects of drought exposure and reflects the main effect. I assume it is not a difference score from the comparison groups.

Thank you for these important questions, which made us realize that the figures require a more detailed explanation. While the regression equation was already included in the Statistical Analysis section, we now clarify in the accompanying text that both the main effects and the interaction terms are included in the model. We also explicitly state that the interaction terms are the primary coefficients of interest already when describing the model. Additionally, we have introduced the term “main effects” at this point, as your comment made us realize that it had previously only appeared in the figure captions. The paragraph now reads as follows (new text in italics):

„DroughtPrenatal is a binary variable describing whether a drought was experienced in utero. DroughtBirthyear refers to drought exposure in the estimated birth year, while Drought123 is a set of dummies indicating drought shocks in the first three years after the birth year. These are the so-called main effects in our model, which are however not our main interest in this study. The β_k are the coefficients of the interaction between prenatal and early-life postnatal drought exposures – the coefficients of interest in this study. X' is a set of controls, which includes the child’s age at assessment, child sex, and a time trend (year of assessment and year of assessment squared), and v is the error term.” (pp. 6- 7)

Since, as you point out, the focus of this study is on the interaction terms testing the PAR hypothesis, you are indeed right that we did not initially emphasize the main effects very much. Your comment showed us that this may cause confusion, as the main effects are also presented in the figures. In response, we have now incorporated the term “main effects” into the section of the results where we describe the overall pattern of findings, and we provide an additional explanation of the main effect results to improve completeness:

“The main effects of prenatal exposure to droughts and test scores in adolescence are negative across all outcome measures. Children who experienced a prenatal drought reach fewer points in total performance (-0.04, 95% CI: -0.07 , -0.01), reading (-0.03, 95% CI: -0.05 , -0.01) and math scores (-0.02, 95% CI: -0.04 , -0.001) than those who did not. Similarly for the binary outcome measures (Panels B and C), main effects of early-life drought exposure are in a negative direction for both the prenatal and early childhood periods, most consistently for the total score and math, with evidence being not as strong as for the linear outcomes.” (p. 9)

Accordingly, we have adjusted the figures to now include the specific wording “main effects” and “interactions”, in accordance with the manuscript text. We hope that in this way, it is now clear which effects are presented where.

3. Children experiencing prenatal or postnatal exposure to drought have lower cognitive scores than comparison children. But those exposed early are not as affected by later drought conditions as prenatally unexposed children. Is this a Floor Effect, or evidence for the Law of Initial Values? That is, are they starting at a lower level so the only way they can move is up?

Thank you for sharing these ideas, which we considered in the context of our study. We describe our results in terms of matching vs mismatching pre- and postnatal circumstances. The results fit to the theory of Predictive Adaptive Responses, and we believe that they cannot be explained by floor effects or the Law of Initial Values.

The reason why we think that our results cannot be due to floor effects is that the mean score is 6.63 out of 8 with a standard deviation of 1.80 (Table 1). This means that there would be plenty of room to go down. Suppose that floor effects would explain our results, then the following scenario would apply.

- The effect of in utero exposure is to reduce cognitive scores by an average of 0.0433 points (Table SI 2).
- This would lead the prenatally exposed to hit a certain bottom.
- And because of this, a postnatal exposure would not impact them so much.

The effect of 0.0433 points, compared to the mean and standard deviation of our dependent variable seems much too small to cause such a floor effect.

The Law of Initial Values is about the level at which an individual starts. Important here is that we are looking at the interaction effect. i.e., we find that the effect of postnatal exposure is less harmful when the individual had already been prenatally exposed. If this would be explainable by the Law of Initial Values, then it would apparently matter whether someone starts with a value that is 0.0433 points lower due to prenatal exposure. Children who were prenatally exposed would then see less impact from a later-life drought because they start this much lower. Again, the prenatal effect compared to the mean and standard deviation of the dependent variable does not seem large enough to cause such a phenomenon.

Beside this quantitative argumentation, it seems to us that the concept of the Law of Initial Values does not fit with what we are studying here. To our knowledge, the Law of Initial Values is often linked to regression to the mean – but we do not see how this would apply in this scenario – or to the tendency of the body to maintain homeostasis. The latter seems to not fit either. Homeostasis is to our understanding not something that applies to cognitive performance, but more to variables like heart rate, blood pressure or hormone levels. If the reviewer means a different process than what we describe here, we are happy to respond to further detail on this.

4. There are other models than PAR, including the Biological Sensitivity to Context and the Stress-Inoculation frameworks. Perhaps the paper could be presented in a broader context, but the review and rationale are both clear and accurate. However, there is a human study with precise predictive and outcome measures that are highly consistent with, and supportive of, the findings in this manuscript. The authors may want to consider if or how the depression exposure in humans in that paper, with the same model, is related to their results. (Psychological Science, 2012, 23 (1), 93-100).

Thank you for pointing us to the paper by Sandman et al. (2012) which we had not been aware of, but which seems very relevant and fitting with our results. The difference is the age of measurement, the type of exposure as well as the sample size, but it provides supportive evidence for the hypothesis we are testing. We now discuss it in our Discussion section:

Moreover, matching vs mismatching pre- and postnatal exposures can also occur with respect to other types of exposures. The literature on this has remained scarce. One of the few available studies is a cohort study on 221 pregnant women in the United States focusing on exposure to

maternal depression [69]. This study – like ours – suggests that matching pre- and postnatal circumstances are beneficial for health: infants had poorer cognitive skills both when maternal depression occurred either prenatally or postnatally. However, when the mother reported depressive symptoms during both periods (i.e., a matching exposure for her child), infant outcomes were comparable to those of infants whose mother never experienced a depression [69]. (p. 11)

We did not pick up the theories on biological sensitivity to context and stress inoculation. According to our reading of the literature, stress inoculation has not been linked to (early-life) nutrition. It appears more commonly referred to in the context of the development of resilience after exposure to stress during postnatal phases. This applies to Biological Sensitivity to Context (BSCT) as well, with the difference that it would lead to the opposite type of pattern than the one we observe. (Following BSCT, early-life exposure to stressors makes individuals more responsive to contexts. This means that prenatally exposed individuals would respond more strongly to later-life adversity. This is the opposite of what we find.) We thus found that both theories do not fit well with the nutritional exposures that we study, and that discussing them might therefore distract from more fitting theories.

References

- Banerji, R., Bhattacharjea, S., & Wadhwa, W. (2013). The annual status of education report (ASER). *Research in Comparative and International Education*, 8(3), 387–396.
- De Onis, M., & Branca, F. (2016). Childhood stunting: a global perspective. *Maternal & child nutrition*, 12, 12–26.
- Guidi, J., Lucente, M., Sonino, N., & Fava, G. A. (2020). Allostatic load and its impact on health: a systematic review. *Psychotherapy and Psychosomatics*, 90(1), 11–27.
- Hawkes, C. H., Del Tredici, K., & Braak, H. (2009). Parkinson's disease: the dual hit theory revisited. *Annals of the New York Academy of Sciences*, 1170(1), 615–622.
- Karimi, S. M., Little, B. B., & Mokhtari, M. (2021). Short-term fetal nutritional stress and long-term health: Child height. *American Journal of Human Biology*, 33(6), e23531.
- Kawano, A., Kelp, M., Qiu, M., Singh, K., Chaturvedi, E., Dahiya, S., Azevedo, I., & Burke, M. (2025). Improved daily PM2.5 estimates in India reveal inequalities in recent enhancement of air quality. *Science Advances*, 11(4), eadq1071.
- Lassi, Z. S., Padhani, Z. A., Salam, R. A., & Bhutta, Z. A. (2022). Prenatal nutrition and nutrition in pregnancy: effects on long-term growth and development. In *Early nutrition and long-term health* (pp. 397–417). Elsevier.
- Lumey, L. H., Stein, A. D., & Susser, E. (2011). Prenatal famine and adult health. *Annual Review of Public Health*, 32(1), 237–262.
- Prendergast, A. J., & Humphrey, J. H. (2014). The stunting syndrome in developing countries. *Paediatrics and international child health*, 34(4), 250–265.
- Sandman, C. A., Davis, E. P., & Glynn, L. M. (2012). Prescient human fetuses thrive. *Psychological Science*, 23(1), 93–100.
- Wells, J. C. (2012). A critical appraisal of the predictive adaptive response hypothesis. *International Journal of Epidemiology*, 41(1), 229–235.
- World Health Organization. (2023). WHO rolls out new holistic way to measure early childhood development. Retrieved 2025/12/19 from <https://www.who.int/news/item/27-02-2023-who-rolls-out-new-holistic-way-to-measure-early-childhood-development>